# Similarity Measure for Interval Neutrosophic Sets and Its Decision Application in Resource Offloading of Edge Computing

**Qiong Liu** [1,2,*] , **Xi Wang** [1,3,*] , **Mingming Kong** [1,3] **and Keyun Qin** [1]

1 School of Computing and Artificial Intelligence, Southwest Jiaotong University, Chengdu 611756, China
2 School of Science, Xihua University, Chengdu 610039, China
3 School of Computer and Software Engineering, Xihua University, Chengdu 610039, China
* Correspondence: liuqiong2013@yeah.net (Q.L.); wangxi_990117@163.com (X.W.)

**Abstract:** Interval neutrosophic sets (INSs), characterized by truth, indeterminacy and falsity membership degrees, handle the uncertain and inconsistent information that commonly exists in real-life systems, and constitute an extension of the interval valued fuzzy set and interval valued intuitionistic fuzzy set. The existing works on similarity measures for INSs are mostly constructed by distance measures and entropies. Meanwhile, the degree of similarity is expressed as a single number, even if the interval-valued information is considered. This may lead to a loss of interval-valued information. In order to cope with these issues, in this paper, we introduce a new approach to constructing the similarity measures for INSs using fuzzy equivalencies. First, based on fuzzy equivalencies and aggregation operators, the definition of interval-valued fuzzy equivalence is generalized to interval neutrosophic values. Then, based on the framework of INSs, we propose the definition and construction method of the similarity measure using the interval neutrosophic fuzzy equivalence. The similarity degree is expressed as an interval and could retain more information than ever before. In addition, according to practical situations, one can obtain different similarities by selecting the parameters in fuzzy equivalence. Due to the increase in edge computing, it is necessary to reasonably offload the client's resource and assign them to the edge server to balance the resource usage. The Similarity measure is conductive to select and match the client and edge server. Finally, an illustrative example verifies that the proposed method can find a reasonable client and edge server, as well as effectiveness in the edge computing application.

**Keywords:** interval neutrosophic sets; similarity measure; resource offloading

## 1. Introduction

Edge computing refers to technologies that allow computing to be performed at the edge of the network on downstream data, meaning cloud services, and upstream data, meaning Internet of Things services [1]. The advantages of edge computing include its ability to overcome the restrictions on limited computation capacity for some clients compared with local computing. At the same time, in contrast with offloading resources toward the remote cloud, edge computing can avoid the high latencies caused by the offloading of certain tasks [2]. Therefore, offloading resources is considered a critical challenge in edge computing. In order to ensure the normal operation of services for clients, redundant work is allocated to edge servers based on the load capacity. This can enhance the speed of response to new services and improve the robustness and computing ability of the network.

In essence, the selection and matching of clients and edge servers for resource offloading are the multi-attribute decision-making (MADM) problems. Due to the ambiguity of people's thinking and the complexity of objective things, the attribute values of MADM problems cannot be expressed by crisp numbers and may be easier to describe by fuzzy

information. Zadeh [3] proposed the theory of fuzzy sets in 1965. The concept of fuzzy sets opened up new perspectives to handle the hesitation and vagueness comprised in the decision-making scheme. This has been studied at length and successfully applied in various fields [4–7]. As the fuzzy set uses one single value $T_A(x) \in [0,1]$ to represent the grade of truth-membership of the fuzzy set $A$ in the universe, it cannot handle some cases where $T_A$ is difficult to define by a specific value, so Turksen [8] proposed the interval valued fuzzy sets. However, whether these are fuzzy sets or interval valued fuzzy sets, it is difficult to describe decision-makers' evaluation of complex objects only through the truth-membership function in practical decision making. On this basis, Atanassov [9,10] proposed intuitionistic fuzzy sets, and added a falsity-membership function to the fuzzy set, which is an extension of Zadeh's fuzzy sets. That is to say that there is a truth-membership function $T_A(x)$ and a falsity-membership function $F_A(x)$ in an intuitionistic fuzzy set $A$, which satisfy the conditions $T_A(x), F_A(x) \in [0,1]$ and $0 \leq T_A(x) + F_A(x) \leq 1$. Furthermore, Atanassov and Gargov [11,12] proposed the interval-valued intuitionistic fuzzy sets by extending the truth-membership function and falsity-membership function to interval values. However, intuitionistic fuzzy sets and interval-valued intuitionistic fuzzy sets can only handle incomplete information, but cannot deal with the uncertain information and inconsistent information in practical decision-making problems. For example, in the voting issue, some agreed, some opposed and some abstained. Another example relates to medical diagnoses: sometimes it is difficult for a doctor to make a certain diagnosis when a patient is suffering from a disease, so they may give an analysis with a degree of truth and falsity, as well as indeterminacy, such as "yes" (60%), "no" (40%) and "not sure" (20%) in [13,14]. These issues are beyond the scope of fuzzy sets and intuitionistic fuzzy sets. Therefore, Smarandache [15] proposed the concept of neutrosophic sets (NSs), which are independently characterized by the truth-membership function $T_A(x)$, the falsity-membership function $F_A(x)$, and the indeterminacy-membership function $I_A(x)$. In the neutrosophic set, the indeterminacy factor is explicitly quantified, and completely independent from the truth-membership and false-membership, while the incorporated uncertainty is dependent on the degrees of belongingness and non-belongingness in the intuitionistic fuzzy set. With regard to the aforementioned example about the doctor's diagnosis, it can be expressed as $(0.6, 0.4, 0.2)$ by NSs. To date, NSs have become an interesting research topic and attracted widespread attention. The original NS is mainly used for philosophical applications: in order to easily use the NS in real scientific and engineering fields, some extensions to NS have been proposed. Wang et al. [16] proposed the notion of a single-valued neutrosophic set (SVNS), which is an instance of NS, and provided a set of theoretic operations on SVNSs. Zhang et al. [17,18] performed extensive research on neutrosophic sets, and proposed a new kind of inclusion relation and new operations in SVNSs. Similarly to interval intuitionistic fuzzy sets, Wang et al. [19] proposed interval neutrosophic sets (INSs), wherein the truth-membership, indeterminacy-membership, and false-membership were extended to interval numbers, and discussed some properties. From a practical point of view, using interval values to express uncertain information is more appropriate than a single value and more suitable for people's needs. As a combination of interval-valued sets and SVNSs, INSs provide an effective approach to deal with uncertain, inconsistent, incomplete, and imprecise information. The INS theory has been proven to be useful in many scientific fields, such as multi-attribute decision making, machine learning, algebraic systems, etc. [20–22].

Information measures are crucial to decision making in uncertain information processing, and similarity is the most important measure. In general, similarity measures are mainly used to measure the discrimination degree of objects. In many practical situations, we need to compare two objects in order to determine whether they are identical or approximately identical or at least to what degree they are identical. To date, a lot of research has been performed on similarity measures in the field of NS theory. Broumi and Smarandacha [23] presented a method to calculate the distance between SVNSs on the basis of the Hausdorff distance and proposed some similarity measures based on the distance and matching function to calculate the similarity degree between SVNSs. Majumdar and

Samanta [24] presented several similarity measures for SVNSs based on the Hamming (Euclidian) distance and normalized Hamming (Euclidian) distance between two SVNSs. Ye [21,25–27] studied the similarity of SVNSs and INSs from different angles. Wang [28] discussed the relationship among several existing similarity measures of SVNSs, such as distance-based similarity measures, the similarity measures based on min and max operators, and vector similarity measures in terms of inequality and equivalence, and provided the definition of equivalence for similarity measures. In addition, Qin [29] proposed a new similarity and entropy based on the new inclusion relationship of SVNSs [17,18]. Yang [30] defined a new inclusion relationship of INSs, and gave the new similarity and entropy for the new inclusion relationship. In accordance with the fact that most cases of similarity among SVNSs are often counter-intuitive, Zeng [31] constructed a new distance measure of SVNSs based on the modified Manhattan distance and proposed a new distance-based similarity measure. Ali [32] developed two forms of Hausdorff distance between SVNSs based on the definition of an Hausdorff metric between two sets, and used these new distance measures to construct several similarity measures for SVNSs. It is easy to ascertain that the similarity measures are expressed by a single number regardless of whether on is dealing with SVNSs or INSs. This is why it is more reasonable to think that interval neutrosophic sets express more uncertain information than single-valued neutrosophic sets, since the interval values could prevent information loss to a maximum extent. However, there has been little research using interval values to express the similarity between INSs to date.

In most research, similarity measures are constructed based on the distance measure and entropy [33–35]. However, as a fuzzy connective, the fuzzy equivalence has been used to depict the similarity between fuzzy sets from another aspect. Fodor and Roubens [36] first put forward the concept of fuzzy equivalence. Wang et al. [37] proposed a way of constructing fuzzy equivalencies using fuzzy implications. Li et al. [38] proved that the biresiduations of $t$ -norms are indeed fuzzy equivalencies, and then presented several ways of constructing fuzzy equivalencies based on the composition of automorphisms, fuzzy negations, and some existing fuzzy equivalencies. Li et al. [39] introduced the concept of an interval-valued fuzzy equivalence, which is an interval extension of the fuzzy equivalence, and provided ways of constructing interval-valued fuzzy equivalencies from given fuzzy equivalencies and aggregation functions. Inspired by the concept of interval-valued fuzzy equivalence, in the present paper, we consider extending the interval-valued fuzzy equivalencies to interval neutrosophic fuzzy equivalencies, and then we propose a new method of similarity measure for INSs based on the fuzzy equivalence and express the similarity with interval values. The advantage of this method is that, on the one hand, different similarities could be obtained by transforming the parameters in fuzzy equivalence, making the similarity methods more inclusive. On the other hand, the similarity degree in interval form could retain more information than ever before.

The rest of this paper is organized as follows. In Section 2, we review some preliminary definitions and the results of the interval neutrosophic set and fuzzy equivalence on the unit interval. In Section 3, based on the interval-valued fuzzy equivalence, we define the concept of an interval neutrosophic fuzzy equivalence, and provide a way of constructing interval neutrosophic fuzzy equivalencies using the aggregation operators. In Section 4, we extend the definition and construction method of a similarity measure for interval neutrosophic values to interval neutrosophic sets. In Section 5, we present a multi-attribute decision-making method using the similarity measure on interval neutrosophic sets, and apply it to the selection and matching of clients and edge servers for resource offloading. A brief conclusion is provided in Section 6.

## 2. Preliminaries

This section gives a brief overview of the concepts of interval-valued sets, interval neutrosophic sets, and the fuzzy equivalence of real numbers on $[0, 1]$.

*2.1. Interval-Valued Set and Its Operational Rules*

Consider the real unit interval $[0,1]$, and denote the set $N_{[0,1]} = \{[a^L, a^U] | 0 \le a^L \le a^U \le 1\}$ as the interval-valued set of the closed interval $[0,1]$. An interval value $a \in N_{[0,1]}$ can be denoted by $a = [a^L, a^U]$, where $a^L$ and $a^U$ are the left and right endpoints of the interval value $a$, respectively.

The usual partial orders between interval values are the product order and the inclusion order. For given interval values $a, b \in N_{[0,1]}$, the product order $\le$ is defined as $a \le b$, if and only if $a^L \le b^L$ and $a^U \le b^U$, and the inclusion order $\subseteq$ is defined as $a \subseteq b$, if and only if $b^L \le a^L$ and $a^U \le b^U$ [39].

According to Zadeh's extension principle, we can extend the logic operation $\vee$, $\wedge$, $^c$ on the closed interval $[0,1]$ to the interval values set $N_{[0,1]}$, then some operations can be presented as follows: for any $a = [a^L, a^U]$, $b = [b^L, b^U] \in N_{[0,1]}$, for any $\lambda \in [0,1]$,

(1)　$a \vee b = [a^L \vee b^L, a^U \vee b^U]$;

(2)　$a \wedge b = [a^L \wedge b^L, a^U \wedge b^U]$;

(3)　$a^c = [1 - a^U, 1 - a^L]$;

(4)　$\min N_{[0,1]} = [0,0]$, $\max N_{[0,1]} = [1,1]$;

(5)　$a + b = [(a^L + b^L) \wedge 1, (a^U + b^U) \wedge 1]$;

(6)　$\lambda a = [\lambda a^L, \lambda a^U]$;

*2.2. Interval Neutrosophic Sets*

The neutrosophic set is a part of neutrosophy, which studies the origin, nature, and scope of neutralities, as well as their interactions with different ideational spectra, and is a powerful general formal framework, which generalizes the aforementioned sets from a philosophical point of view. Smarandache [15] gave the following definition of a neutrosophic set.

**Definition 1** ([15])**.** *Let X be a space of points (objects), with a generic element in X denoted by x. A neutrosophic set A in X is characterized by a truth-membership function $T_A(x)$, an indeterminacy-membership function $I_A(x)$, and a falsity-membership function $F_A(x)$, where $T_A(x)$, $I_A(x)$, and $F_A(x)$ are real standard or nonstandard subsets of $]0^-, 1^+[$, such that $T_A(x) : X \to ]0^-, 1^+[$, $I_A(x) : X \to ]0^-, 1^+[$ and $F_A(x) : X \to ]0^-, 1^+[$. Furthermore, the sum of $T_A(x)$, $I_A(x)$, and $F_A(x)$ satisfies the condition $0^- \le \sup T_A(x) + \sup I_A(x) + \sup F_A(x) \le 3^+$.*

An interval neutrosophic set is an instance of neutrosophic set which can be used in real scientific and engineering applications. In the following, we introduce the definition of an interval neutrosophic set.

**Definition 2** ([19])**.** *Let X be a space of points (objects), with a generic element in X denoted by x. An interval neutrosophic set A in X is characterized by a truth-membership degree $T_A(x)$, an indeterminacy-membership degree $I_A(x)$, and a falsity-membership degree $F_A(x)$. An interval neutrosophic set A can be denoted by*

$$A = \{< x, (T_A(x), I_A(x), F_A(x)) > | x \in X\}, \tag{1}$$

*where $T_A(x), I_A(x), F_A(x) \subseteq [0,1]$ for each $x \in X$, and the sum of supremum of $T_A(x)$, $I_A(x)$ and $F_A(x)$ satisfies the condition $0 \le \sup T_A(x) + \sup I_A(x) + \sup F_A(x) \le 3$.*

For convenience, if $T_A(x) = [T_A^L(x), T_A^U(x)]$, $I_A(x) = [I_A^L(x), I_A^U(x)]$ and $F_A(x) = [F_A^L(x), F_A^U(x)]$, then

$$A = \{< x, ([T_A^L(x), T_A^U(x)], [I_A^L(x), I_A^U(x)], [F_A^L(x), F_A^U(x)]) > | x \in X\}. \tag{2}$$

We use the symbol $INS(X)$ to denote the set of all interval neutrosophic sets in $X$.

**Definition 3** ([19]). *Let $X$ be a finite set and $A, B \in INS(X)$. A is contained in B, denoted by $A \subseteq B$, if $T_A^L(x) \leq T_B^L(x)$, $T_A^U(x) \leq T_B^U(x)$, $I_A^L(x) \geq I_B^L(x)$, $I_A^U(x) \geq I_B^U(x)$, and $F_A^L(x) \geq F_B^L(x)$, $F_A^U(x) \geq F_B^U(x)$ for any $x \in X$.*

**Definition 4** ([19]). *Let A be an interval neutrosophic set in X,*

*(1) If $T_A^L(x) = T_A^U(x) = 0$, $I_A^L(x) = I_A^U(x) = 1$, $F_A^L(x) = F_A^U(x) = 1$ for all $x \in X$, then A is called a null interval neutrosophic set, denoted by $\varnothing_{INS}$;*

*(2) If $T_A^L(x) = T_A^U(x) = 1$, $I_A^L(x) = I_A^U(x) = 0$, $F_A^L(x) = F_A^U(x) = 0$ for all $x \in X$, then A is called an absolute interval neutrosophic set, denoted by $U_{INS}$;*

For the convenience of discussion on the interval neutrosophic set, we introduce the concept of interval neutrosophic value [17,30]. Let the set

$$\begin{aligned} \widetilde{D}^* &= \{\widetilde{x} = (\widetilde{x}_1, \widetilde{x}_2, \widetilde{x}_3) | \widetilde{x}_1, \widetilde{x}_2, \widetilde{x}_3 \subseteq [0,1]\} \\ &= \{\widetilde{x} = ([x_1^L, x_1^U], [x_2^L, x_2^U], [x_3^L, x_3^U]) | [x_i^L, x_i^U] \subseteq [0,1], i = 1,2,3\}. \end{aligned} \tag{3}$$

For $i = 1, 2, 3$, $\widetilde{x}_i = [x_i^L, x_i^U]$ be a subinterval of $[0,1]$. An element of $\widetilde{D}^*$ is called an interval neutrosophic value. For any interval neutrosophic set $A$, $(T_A(x), I_A(x), F_A(x)) \in \widetilde{D}^*$.

Associated with the inclusion relation $\subseteq$ of interval neutrosophic sets on Definition 3, we can give the order relation on $\widetilde{D}^*$, which is based on the product order $\leq$ of the interval-valued set. For any $\widetilde{x} = (\widetilde{x}_1, \widetilde{x}_2, \widetilde{x}_3)$, $\widetilde{y} = (\widetilde{y}_1, \widetilde{y}_2, \widetilde{y}_3) \in \widetilde{D}^*$, $\widetilde{x} \leq \widetilde{y} \Leftrightarrow (\widetilde{x}_1 \leq \widetilde{y}_1) \wedge (\widetilde{x}_2 \geq \widetilde{y}_2) \wedge (\widetilde{x}_3 \geq \widetilde{y}_3)$.

*2.3. Fuzzy Equivalence and Aggregation Function*

**Definition 5** ([36,38,39]). *A function $E : [0,1] \times [0,1] \to [0,1]$ is called a fuzzy equivalence if it satisfies the following properties:*

*(E1) $E(x,y) = E(y,x)$ for any $x, y \in [0,1]$;*

*(E2) $E(x,x) = 1$ for any $x \in [0,1]$;*

*(E3) $E(1,0) = E(0,1) = 0$;*

*(E4) For all $x, y, z \in [0,1]$, if $x \leq y \leq z$, then $E(x,z) \leq \min\{E(x,y), E(y,z)\}$;*

**Example 1** ([38]). *$E_{(\theta,\varepsilon)}^{\alpha}$ and $E_{(\theta,\varepsilon)}^{\beta}$ are two general forms of fuzzy equivalence for any $x, y \in [0,1]$, respectively, given by:*

$$E_{(\theta,\varepsilon)}^{\alpha}(x,y) = \frac{\theta - \theta|x-y| + \varepsilon \min(x,y)}{\theta - (\theta-1)|x-y| + \varepsilon \min(x,y)}, \tag{4}$$

$$E_{(\theta,\varepsilon)}^{\beta}(x,y) = \frac{\theta - \theta|x-y| + \varepsilon \min(1-x, 1-y)}{\theta - (\theta-1)|x-y| + \varepsilon \min(1-x, 1-y)}, \tag{5}$$

*with $\theta \geq 0$, $\varepsilon \geq 0$. Furthermore, we can give the particular values of the parameter $\theta$ and $\varepsilon$, and then some specific fuzzy equivalencies can be generated by the above formulas.*

*(1) If $\theta = 0$, $\varepsilon = 1$, then we have*

$$E_{(0,1)}^{\alpha}(x,y) = \frac{\min(x,y)}{|x-y| + \min(x,y)} = \frac{\min(x,y)}{\max(x,y)} = \frac{x \wedge y}{x \vee y}.$$

$$E_{(0,1)}^{\beta}(x,y) = \frac{\min(1-x, 1-y)}{|x-y| + \min(1-x, 1-y)} = \frac{\min(1-x, 1-y)}{\max(1-x, 1-y)} = \frac{(1-x) \wedge (1-y)}{(1-x) \vee (1-y)}.$$

*(2) If $\theta = 0$, $\varepsilon = 2$, then we have*

$$E^{\alpha}_{(0,2)}(x,y) = \frac{2\min(x,y)}{x+y} = \frac{2(x \wedge y)}{x+y}.$$

$$E^{\beta}_{(0,2)}(x,y) = \frac{2\min(1-x,1-y)}{|x-y|+2\min(1-x,1-y)} = \frac{2\min(1-x,1-y)}{2-x-y)} = \frac{2(1-x) \wedge (1-y)}{2-x-y}.$$

*(3) If $\theta = 1$, $\varepsilon = 0$, then we have*

$$E^{\alpha}_{(1,0)}(x,y) = E^{\beta}_{(1,0)}(x,y) = 1 - |x-y|.$$

**Definition 6** ([38,40,41]). *A function $Ag : [0,1]^n \rightarrow [0,1]$ is an aggregation operator if it satisfies the following properties:*

*(Ag1) $Ag(x,x,\ldots,x) = x$ for all $x \in [0,1]$.*

*(Ag2) $Ag$ is monotonically increasing in all of its arguments.*

**Example 2** ([38,40]). *We can take some examples of aggregation operator as follows:*

*(1) The arithmetic mean: $Ag_{a-mean}(x_1,x_2,\ldots,x_n) = \frac{1}{n}\sum_{i=1} x_i$.*

*(2) The geometric mean: $Ag_{g-mean}(x_1,x_2,\ldots,x_n) = \sqrt[n]{\prod_{i=1}^{n} x_i}$.*
*Especially, for any $x,y \in [0,1]$, $Ag_{a-mean}(x,y) = \frac{x+y}{2}$, and $Ag_{g-mean}(x,y) = \sqrt{xy}$. It is easy to know that for any $x,y \in [0,1]$, $Ag_{g-mean}(x,y) \leq Ag_{a-mean}(x,y)$.*

*(3) The convex linear combinations:*
*$Ag_{\lambda}(x_1,x_2,\ldots,x_n) = \lambda\min(x_1,x_2,\ldots,x_n) + (1-\lambda)\max(x_1,x_2,\ldots,x_n)$, with $\lambda \in [0,1]$.*
*Especially, for any $x,y \in [0,1]$, $Ag_{\lambda}(x,y) = \lambda\min(x,y) + (1-\lambda)\max(x,y)$. In particular, if $\lambda = 0$, we have $Ag_0(x,y) = max(x,y) = x \vee y$; if $\lambda = 1$, we have $Ag_1(x,y) = min(x,y) = x \wedge y$. Moreover, for any $\lambda_1,\lambda_2 \in [0,1]$, $\lambda_1 < \lambda_2$, then $Ag_{\lambda_2}(x,y) \leq Ag_{\lambda_1}(x,y)$.*

## 3. Fuzzy Equivalence on Interval Neutrosophic Values

In fuzzy set theory, fuzzy equivalences can be used to construct similarity measures for fuzzy sets. In order to describe the similarity of interval neutrosophic sets, in this section, we will introduce the concept of an interval neutrosophic valued fuzzy equivalence. Li [39] introduced the concept of interval-valued fuzzy equivalence.

**Definition 7** ([39]). *A function $IE : N^2_{[0,1]} \rightarrow N_{[0,1]}$ is called an interval-valued fuzzy equivalence, if it satisfies the following properties:*

*(IE1) $IE(a,b) = IE(b,a)$ for all $a,b \in N_{[0,1]}$;*

*(IE2) $IE(a,a) = [1,1]$ for any $a \in N_{[0,1]}$;*

*(IE3) $IE([0,0],[1,1]) = IE([1,1],[0,0]) = [0,0]$;*

*(IE4) For all $a,b,c \in N_{[0,1]}$, and if $a \leq b \leq c$, then $IE(a,c) \leq IE(a,b), IE(a,c) \leq IE(b,c)$.*

Based on the given fuzzy equivalencies and aggregation functions, we can construct a general formalization for an interval-valued fuzzy equivalence [39].

**Theorem 1** ([39]). *Let $E^i_{(\theta,\varepsilon)}$ $(i = \alpha, \beta)$ be the given fuzzy equivalencies; and let $Ag_\mu$ and $Ag_\varphi$ be two aggregation functions such that $Ag_\mu \leq Ag_\varphi$. Then, the function $IE^i_{(\mu,\varphi),(\theta,\varepsilon)} : N^2_{[0,1]} \rightarrow N_{[0,1]}$ $(i = \alpha, \beta)$ is an interval-valued fuzzy equivalence, and it is defined as follows, for any $a,b \in N_{[0,1]}$,*

$$IE^{\alpha}_{(\mu,\varphi),(\theta,\varepsilon)}(a,b) = [Ag_\mu(E^{\alpha}_{(\theta,\varepsilon)}(a^L,b^L), E^{\alpha}_{(\theta,\varepsilon)}(a^U,b^U)), Ag_\varphi(E^{\alpha}_{(\theta,\varepsilon)}(a^L,b^L), E^{\alpha}_{(\theta,\varepsilon)}(a^U,b^U))], \tag{6}$$

$$IE^{\beta}_{(\mu,\varphi),(\theta,\varepsilon)}(a,b) = [Ag_{\mu}(E^{\beta}_{(\theta,\varepsilon)}(a^L,b^L), E^{\beta}_{(\theta,\varepsilon)}(a^U,b^U)), Ag_{\varphi}(E^{\beta}_{(\theta,\varepsilon)}(a^L,b^L), E^{\beta}_{(\theta,\varepsilon)}(a^U,b^U))]. \tag{7}$$

**Proof.** Since $Ag_{\mu} \leq Ag_{\varphi}$, then for $i = \alpha, \beta$, we have $IE^i_{(\mu,\varphi),(\theta,\varepsilon)} \in N_{[0,1]}$. The following items show that $IE^i_{(\mu,\varphi),(\theta,\varepsilon)}$ satisfies the four properties of Definition 7.

(1) For $i = \alpha, \beta$, since $E^i_{(\theta,\varepsilon)}$ is a fuzzy equivalence, then $E^i_{(\theta,\varepsilon)}$ should satisfy the symmetry, and thus, we have $IE^i_{(\mu,\varphi),(\theta,\varepsilon)}(a,b) = IE^i_{(\mu,\varphi),(\theta,\varepsilon)}(b,a)$.

(2) For any $x \in [0,1]$ and $i = \alpha, \beta$, we have $E^i_{(\theta,\varepsilon)}(x,x) = 1$. Meanwhile, for the aggregation functions $Ag_{\mu}$ and $Ag_{\varphi}$, it is clear that $Ag_{\mu}(1,1) = 1$ and $Ag_{\varphi}(1,1) = 1$, thus $IE^i_{(\mu,\varphi),(\theta,\varepsilon)}(a,a) = [1,1]$ for any $a \in N_{[0,1]}$.

(3) For $i = \alpha, \beta$, $E^i_{(\theta,\varepsilon)}(1,0) = E^i_{(\theta,\varepsilon)}(0,1) = 0$. Furthermore, the aggregation function $Ag_{\mu}(0,0) = 0$ and $Ag_{\varphi}(0,0) = 0$, thus $IE^i_{(\mu,\varphi),(\theta,\varepsilon)}([1,1],[0,0]) = IE^i_{(\mu,\varphi),(\theta,\varepsilon)}([0,0],[1,1]) = [0,0]$.

(4) For all $a,b,c \in N_{[0,1]}$, if $a \leq b \leq c$, then $a^L \leq b^L \leq c^L$, and $a^U \leq b^U \leq c^U$.

For $i = \alpha, \beta$, on the one hand, we have $E^i_{(\theta,\varepsilon)}(a^L,c^L) \leq E^i_{(\theta,\varepsilon)}(a^L,b^L)$, and $E^i_{(\theta,\varepsilon)}(a^U,c^U) \leq E^i_{(\theta,\varepsilon)}(a^U,b^U)$. Furthermore, due to monotonic increase in the aggregation function $Ag$, that is, for any $x_1, x_2, y_1, y_2 \in [0,1]$, $Ag(x_1,x_2) \leq Ag(y_1,y_2)$ whenever $x_1 \leq y_1$ and $x_2 \leq y_2$. Then, $Ag_{\mu}(E^i_{(\theta,\varepsilon)}(a^L,c^L), E^i_{(\theta,\varepsilon)}(a^U,c^U)) \leq Ag_{\mu}(E^i_{(\theta,\varepsilon)}(a^L,b^L), E^i_{(\theta,\varepsilon)}(a^U,b^U))$, and $Ag_{\varphi}(E^i_{(\theta,\varepsilon)}(a^L,c^L), E^i_{(\theta,\varepsilon)}(a^U,c^U)) \leq Ag_{\varphi}(E^i_{(\theta,\varepsilon)}(a^L,b^L), E^i_{(\theta,\varepsilon)}(a^U,b^U))$. Therefore, we have $IE^i_{(\mu,\varphi),(\theta,\varepsilon)}(a,c) \leq IE^i_{(\mu,\varphi),(\theta,\varepsilon)}(a,b)$. On the other hand, we could similarly prove that $IE^i_{(\mu,\varphi),(\theta,\varepsilon)}(a,c) \leq IE^i_{(\mu,\varphi),(\theta,\varepsilon)}(b,c)$. $\square$

According to the given examples of Definitions 5 and 6, we present several computational formula for interval-valued fuzzy equivalence $IE$.

1. For $0 \leq \lambda_1 < \lambda_2 \leq 1$, suppose that $Ag_{\mu} = Ag_{\lambda_2}$, $Ag_{\varphi} = Ag_{\lambda_1}$,

$$IE^{\alpha}_{(\lambda_2,\lambda_1),(\theta,\varepsilon)}(a,b) = [Ag_{\lambda_2}(E^{\alpha}_{(\theta,\varepsilon)}(a^L,b^L), E^{\alpha}_{(\theta,\varepsilon)}(a^U,b^U)), Ag_{\lambda_1}(E^{\alpha}_{(\theta,\varepsilon)}(a^L,b^L), E^{\alpha}_{(\theta,\varepsilon)}(a^U,b^U))], \tag{8}$$

$$IE^{\beta}_{(\lambda_2,\lambda_1),(\theta,\varepsilon)}(a,b) = [Ag_{\lambda_2}(E^{\beta}_{(\theta,\varepsilon)}(a^L,b^L), E^{\beta}_{(\theta,\varepsilon)}(a^U,b^U)), Ag_{\lambda_1}(E^{\beta}_{(\theta,\varepsilon)}(a^L,b^L), E^{\beta}_{(\theta,\varepsilon)}(a^U,b^U))]. \tag{9}$$

Importantly, if $\lambda_2 = 1$, $\lambda_1 = 0$, $\theta = 0$, $\varepsilon = 1$, by Equations (8) and (9), we have

$$IE^{\alpha}_{(1,0),(0,1)}(a,b) = [Ag_1(E^{\alpha}_{(0,1)}(a^L,b^L), E^{\alpha}_{(0,1)}(a^U,b^U)), Ag_0(E^{\alpha}_{(0,1)}(a^L,b^L), E^{\alpha}_{(0,1)}(a^U,b^U))]$$

$$IE^{\beta}_{(1,0),(0,1)}(a,b) = [Ag_1(E^{\beta}_{(0,1)}(a^L,b^L), E^{\beta}_{(0,1)}(a^U,b^U)), Ag_0(E^{\beta}_{(0,1)}(a^L,b^L), E^{\beta}_{(0,1)}(a^U,b^U))]$$

Due to the aggregation operators $Ag_0(x,y) = x \vee y$ and $Ag_1(x,y) = x \wedge y$ of Example 2, for $E^{\alpha}_{(0,1)}(x,y) = \frac{x \wedge y}{x \vee y}$ of Example 1, then

$$IE^{\alpha}_{(1,0),(0,1)}(a,b) = [Ag_1(\frac{a^L \wedge b^L}{a^L \vee b^L}, \frac{a^U \wedge b^U}{a^U \vee b^U}), Ag_0(\frac{a^L \wedge b^L}{a^L \vee b^L}, \frac{a^U \wedge b^U}{a^U \vee b^U})]$$
$$= [\frac{a^L \wedge b^L}{a^L \vee b^L} \wedge \frac{a^U \wedge b^U}{a^U \vee b^U}, \frac{a^L \wedge b^L}{a^L \vee b^L} \vee \frac{a^U \wedge b^U}{a^U \vee b^U}], \tag{10}$$

for $E^{\beta}_{(0,1)}(x,y) = \frac{(1-x)\wedge(1-y)}{(1-x)\vee(1-y)}$ of Example 1, then

$$
\begin{aligned}
IE^{\beta}_{(1,0),(0,1)}(a,b) = &\left[Ag_1\left(\frac{(1-a^L)\wedge(1-b^L)}{(1-a^L)\vee(1-b^L)}, \frac{(1-a^U)\wedge(1-b^U)}{(1-a^U)\vee(1-b^U)}\right),\right.\\
&\left.Ag_0\left(\frac{(1-a^L)\wedge(1-b^L)}{(1-a^L)\vee(1-b^L)}, \frac{(1-a^U)\wedge(1-b^U)}{(1-a^U)\vee(1-b^U)}\right)\right]\\
= &\left[\frac{(1-a^L)\wedge(1-b^L)}{(1-a^L)\vee(1-b^L)} \wedge \frac{(1-a^U)\wedge(1-b^U)}{(1-a^U)\vee(1-b^U)},\right.\\
&\left.\frac{(1-a^L)\wedge(1-b^L)}{(1-a^L)\vee(1-b^L)} \vee \frac{(1-a^U)\wedge(1-b^U)}{(1-a^U)\vee(1-b^U)}\right],
\end{aligned}
\tag{11}
$$

Similarly, if $\lambda_2 = 1$, $\lambda_1 = 0$, $\theta = 0$, $\varepsilon = 2$, by Equations (8) and (9), we have

$$
IE^{\alpha}_{(1,0),(0,2)}(a,b) = \left[\frac{2(a^L\wedge b^L)}{a^L+b^L} \wedge \frac{2(a^U\wedge b^U)}{a^U+b^U}, \frac{2(a^L\wedge b^L)}{a^L+b^L} \vee \frac{2(a^U\wedge b^U)}{a^U+b^U}\right],
\tag{12}
$$

$$
\begin{aligned}
IE^{\beta}_{(1,0),(0,2)}(a,b) = &\left[\frac{2((1-a^L)\wedge(1-b^L))}{2-a^L-b^L} \wedge \frac{2((1-a^U)\wedge(1-b^U))}{2-a^U-b^U},\right.\\
&\left.\frac{2((1-a^L)\wedge(1-b^L))}{2-a^L-b^L} \vee \frac{2((1-a^U)\wedge(1-b^U))}{2-a^U-b^U}\right].
\end{aligned}
\tag{13}
$$

2. Suppose that $Ag_\mu = Ag_{g-mean}$, $Ag_\varphi = Ag_{a-mean}$,

$$
IE^{\alpha}_{(g-mean,a-mean),(\theta,\varepsilon)}(a,b) = \left[\sqrt{E^{\alpha}_{(\theta,\varepsilon)}(a^L,b^L)\cdot E^{\alpha}_{(\theta,\varepsilon)}(a^U,b^U)}, \frac{E^{\alpha}_{(\theta,\varepsilon)}(a^L,b^L)+E^{\alpha}_{(\theta,\varepsilon)}(a^U,b^U)}{2}\right],
\tag{14}
$$

$$
IE^{\beta}_{(g-mean,a-mean),(\theta,\varepsilon)}(a,b) = \left[\sqrt{E^{\beta}_{(\theta,\varepsilon)}(a^L,b^L\cdot E^{\beta}_{(\theta,\varepsilon)}(a^U,b^U)}, \frac{E^{\beta}_{(\theta,\varepsilon)}(a^L,b^L)+E^{\beta}_{(\theta,\varepsilon)}(a^U,b^U)}{2}\right].
\tag{15}
$$

Importantly, if $\mu = g-mean$, $\varphi = a-mean$, $\theta = 0$, $\varepsilon = 1$, by Equations (14) and (15), we have

$$
IE^{\alpha}_{(g-mean,a-mean),(0,1)}(a,b) = \left[\sqrt{\frac{a^L\wedge b^L}{a^L\vee b^L}\cdot\frac{a^U\wedge b^U}{a^U\vee b^U}}, \frac{\frac{a^L\wedge b^L}{a^L\vee b^L}+\frac{a^U\wedge b^U}{a^U\vee b^U}}{2}\right],
\tag{16}
$$

$$
\begin{aligned}
IE^{\beta}_{(g-mean,a-mean),(0,1)}(a,b) = &\left[\sqrt{\frac{(1-a^L)\wedge(1-b^L)}{(1-a^L)\vee(1-b^L)}\cdot\frac{(1-a^U)\wedge(1-b^U)}{(1-a^U)\vee(1-b^U)}},\right.\\
&\left.\frac{\frac{(1-a^L)\wedge(1-b^L)}{(1-a^L)\vee(1-b^L)}+\frac{(1-a^U)\wedge(1-b^U)}{(1-a^U)\vee(1-b^U)}}{2}\right],
\end{aligned}
\tag{17}
$$

if $\mu = g-mean$, $\varphi = a-mean$, $\theta = 0$, $\varepsilon = 2$, by Equations (14) and (15), we have

$$
IE^{\alpha}_{(g-mean,a-mean),(0,2)}(a,b) = \left[\sqrt{\frac{2(a^L\wedge b^L)}{a^L+b^L}\cdot\frac{2(a^U\wedge b^U)}{a^U+b^U}}, \frac{\frac{2(a^L\wedge b^L)}{a^L+b^L}+\frac{2(a^U\wedge b^U)}{a^U+b^U}}{2}\right],
\tag{18}
$$

$$
\begin{aligned}
IE^{\beta}_{(g-mean,a-mean),(0,2)}(a,b) = &\left[\sqrt{\frac{2((1-a^L)\wedge(1-b^L))}{2-a^L-b^L}\cdot\frac{2((1-a^U)\wedge(1-b^U))}{2-a^U-b^U}},\right.\\
&\left.\frac{\frac{2((1-a^L)\wedge(1-b^L))}{2-a^L-b^L}+\frac{2((1-a^U)\wedge(1-b^U))}{2-a^U-b^U}}{2}\right].
\end{aligned}
\tag{19}
$$

For any $\widetilde{x} = (\widetilde{x}_1, \widetilde{x}_2, \widetilde{x}_3) \in \widetilde{D}^*$, because of each component $\widetilde{x}_i \in N_{[0,1]}$, therefore, we consider to extend the interval-valued fuzzy equivalence to interval neutrosophic value.

**Definition 8.** *A function* $INE : \widetilde{D}^* \times \widetilde{D}^* \rightarrow N_{[0,1]}$ *is called an interval neutrosophic fuzzy equivalence, if it satisfies the following properties:*

(INE1) $INE(\widetilde{x}, \widetilde{y}) = INE(\widetilde{y}, \widetilde{x})$ *for all* $\widetilde{x}, \widetilde{y} \in \widetilde{D}^*$;

(INE2) $INE(\widetilde{x}, \widetilde{x}) = [1, 1]$ *for all* $\widetilde{x} \in \widetilde{D}^*$;

(INE3) $INE(\widetilde{D}^*_+, \widetilde{D}^*_-) = INE(\widetilde{D}^*_-, \widetilde{D}^*_+) = [0, 0]$, *where* $\widetilde{D}^*_+$ *and* $\widetilde{D}^*_-$ *are called the positive and negative ideal interval neutrosophic value on* $\widetilde{D}^*$, *respectively, i.e.,* $\widetilde{D}^*_+ = ([1,1], [0,0], [0,0])$ *and* $\widetilde{D}^*_- = ([0,0], [1,1], [1,1])$;

(INE4) *For any* $\widetilde{x}, \widetilde{y}, \widetilde{z} \in \widetilde{D}^*$, *if* $\widetilde{x} \leq \widetilde{y} \leq \widetilde{z}$, *then* $INE(\widetilde{x}, \widetilde{z}) \leq INE(\widetilde{x}, \widetilde{y})$, $INE(\widetilde{x}, \widetilde{z}) \leq INE(\widetilde{y}, \widetilde{z})$.

**Theorem 2.** *Suppose that* $IE_\gamma$, $IE_\xi$, *and* $IE_\eta$ *are interval-valued fuzzy equivalences, a function* $\widetilde{F} : \widetilde{D}^* \times \widetilde{D}^* \rightarrow N_{[0,1]}$ *is defined for all* $\widetilde{x} = (\widetilde{x}_1, \widetilde{x}_2, \widetilde{x}_3), \widetilde{y} = (\widetilde{y}_1, \widetilde{y}_2, \widetilde{y}_3) \in \widetilde{D}^*$ *by:*

$$\widetilde{F}(\widetilde{x}, \widetilde{y}) = \omega_1 IE_\gamma(\widetilde{x}_1, \widetilde{y}_1) + \omega_2 IE_\xi(\widetilde{x}_2, \widetilde{y}_2) + \omega_3 IE_\eta(\widetilde{x}_3, \widetilde{y}_3), \tag{20}$$

*where* $\omega_1, \omega_2, \omega_3 \in [0,1]$ *and* $\omega_1 + \omega_2 + \omega_3 = 1$. *Then,* $\widetilde{F}$ *is interval neutrosophic fuzzy equivalence on* $\widetilde{D}^*$.

**Proof.** Since $IE_\gamma$, $IE_\xi$, and $IE_\eta$ are interval-valued fuzzy equivalences, then $IE_k(k = \gamma, \xi, \eta)$ must satisfy the total properties of Definition 7.

(1) Suppose that $\widetilde{x} = (\widetilde{x}_1, \widetilde{x}_2, \widetilde{x}_3), \widetilde{y} = (\widetilde{y}_1, \widetilde{y}_2, \widetilde{y}_3) \in \widetilde{D}^*$, then $IE_k(\widetilde{x}_i, \widetilde{y}_i) = IE_k(\widetilde{y}_i, \widetilde{x}_i)$ for any $i = 1, 2, 3$ and $k = \gamma, \xi, \eta$, thus $\widetilde{F}(\widetilde{x}, \widetilde{y}) = \widetilde{F}(\widetilde{y}, \widetilde{x})$.

(2) Suppose that $\widetilde{x} = (\widetilde{x}_1, \widetilde{x}_2, \widetilde{x}_3) \in \widetilde{D}^*$, then $IE_k(\widetilde{x}_i, \widetilde{x}_i) = [1, 1]$ for any $i = 1, 2, 3$ and $k = \gamma, \xi, \eta$, thus $\widetilde{F}(\widetilde{x}, \widetilde{x}) = \omega_1 IE_\gamma(\widetilde{x}_1, \widetilde{x}_1) + \omega_2 IE_\xi(\widetilde{x}_2, \widetilde{x}_2) + \omega_3 IE_\eta(\widetilde{x}_3, \widetilde{x}_3) = [1, 1]$.

(3) Since $IE_k([0,0], [1,1]) = IE_k([1,1], [0,0]) = [0,0]$ for $k = \gamma, \xi, \eta$, then $\widetilde{F}(([1,1], [0,0], [0,0]), ([0,0], [1,1], [1,1])) = \omega_1 IE_\gamma([1,1], [0,0]) + \omega_2 IE_\xi([0,0], [1,1]) + \omega_3 IE_\eta([0,0], [1,1]) = [0,0]$, and $\widetilde{F}(([0,0], [1,1], [1,1]), ([1,1], [0,0], [0,0])) = \omega_1 IE_\gamma([0,0], [1,1]) + \omega_2 IE_\xi([1,1], [0,0]) + \omega_3 IE_\eta([1,1], [0,0]) = [0,0]$.

(4) Suppose that $\widetilde{x}, \widetilde{y}, \widetilde{z} \in \widetilde{D}^*$ and $\widetilde{x} \leq \widetilde{y} \leq \widetilde{z}$. According to the order relation of interval neutrosophic values, it follows that $\widetilde{x}_1 \leq \widetilde{y}_1 \leq \widetilde{z}_1$, $\widetilde{z}_2 \leq \widetilde{y}_2 \leq \widetilde{x}_2$, and $\widetilde{z}_3 \leq \widetilde{y}_3 \leq \widetilde{x}_3$. Since $IE_\gamma(\widetilde{x}_1, \widetilde{z}_1) \leq IE_\gamma(\widetilde{x}_1, \widetilde{y}_1)$, $IE_\xi(\widetilde{x}_2, \widetilde{z}_2) \leq IE_\xi(\widetilde{x}_2, \widetilde{y}_2)$, and $IE_\eta(\widetilde{x}_3, \widetilde{z}_3) \leq IE_\eta(\widetilde{x}_3, \widetilde{y}_3)$, then $\widetilde{F}(\widetilde{x}, \widetilde{z}) \leq \widetilde{F}(\widetilde{x}, \widetilde{y})$. On the other hand, since that $IE_\gamma(\widetilde{x}_1, \widetilde{z}_1) \leq IE_\gamma(\widetilde{y}_1, \widetilde{z}_1)$, $IE_\xi(\widetilde{x}_2, \widetilde{z}_2) \leq IE_\xi(\widetilde{y}_2, \widetilde{z}_2)$, and $IE_\eta(\widetilde{x}_3, \widetilde{z}_3) \leq IE_\eta(\widetilde{y}_3, \widetilde{z}_3)$, we thus have $\widetilde{F}(\widetilde{x}, \widetilde{z}) \leq \widetilde{F}(\widetilde{y}, \widetilde{z})$. □

Suppose that, in Theorem 2, we take $\omega_1 = \omega_2 = \omega_3 = \frac{1}{3}$ and $IE_\gamma = IE_\xi = IE_\eta = IE^\alpha_{(\mu,\varphi),(\theta,\varepsilon)}$ (or $IE^\beta_{(\mu,\varphi),(\theta,\varepsilon)}$) and then, based on the above interval-valued fuzzy equivalences, we obtain the corresponding interval neutrosophic fuzzy equivalencies $INE^\alpha_{(\mu,\varphi),(\theta,\varepsilon)}$ and $INE^\beta_{(\mu,\varphi),(\theta,\varepsilon)}$ as follows:

$$INE^\alpha_{(1,0),(0,1)}(\widetilde{x}, \widetilde{y}) = \frac{1}{3} \sum_{i=1}^3 [\frac{x_i^L \wedge y_i^L}{x_i^L \vee y_i^L} \wedge \frac{x_i^U \wedge y_i^U}{x_i^U \vee y_i^U}, \frac{x_i^L \wedge y_i^L}{x_i^L \vee y_i^L} \vee \frac{x_i^U \wedge y_i^U}{x_i^U \vee y_i^U}] \tag{21}$$

$$\begin{aligned} INE^\beta_{(1,0),(0,1)}(\widetilde{x}, \widetilde{y}) = \frac{1}{3} \sum_{i=1}^3 [&\frac{(1 - x_i^L) \wedge (1 - y_i^L)}{(1 - x_i^L) \vee (1 - y_i^L)} \wedge \frac{(1 - x_i^U) \wedge (1 - y_i^U)}{(1 - x_i^U) \vee (1 - y_i^U)}, \\ &\frac{(1 - x_i^L) \wedge (1 - y_i^L)}{(1 - x_i^L) \vee (1 - y_i^L)} \vee \frac{(1 - x_i^U) \wedge (1 - y_i^U)}{(1 - x_i^U) \vee (1 - y_i^U)}] \end{aligned} \tag{22}$$

$$INE^\alpha_{(1,0),(0,2)}(\widetilde{x},\widetilde{y}) = \frac{1}{3}\sum_{i=1}^{3}[\frac{2(x_i^L \wedge y_i^L)}{x_i^L + y_i^L} \wedge \frac{2(x_i^U \wedge y_i^U)}{x_i^U + y_i^U}, \frac{2(x_i^L \wedge y_i^L)}{x_i^L + y_i^L} \vee \frac{2(x_i^U \wedge y_i^U)}{x_i^U + y_i^U}] \quad (23)$$

$$INE^\alpha_{(g-mean,a-mean),(0,1)}(\widetilde{x},\widetilde{y}) = \frac{1}{3}\sum_{i=1}^{3}[\sqrt{\frac{x_i^L \wedge y_i^L}{x_i^L \vee y_i^L} \cdot \frac{x_i^U \wedge y_i^U}{x_i^U \vee y_i^U}}, \frac{\frac{x_i^L \wedge y_i^L}{x_i^L \vee y_i^L} + \frac{x_i^U \wedge y_i^U}{x_i^U \vee y_i^U}}{2}] \quad (24)$$

$$INE^\beta_{(g-mean,a-mean),(0,1)}(\widetilde{x},\widetilde{y}) = \frac{1}{3}\sum_{i=1}^{3}[\sqrt{\frac{(1-x_i^L) \wedge (1-y_i^L)}{(1-x_i^L) \vee (1-y_i^L)} \cdot \frac{(1-x_i^U) \wedge (1-y_i^U)}{(1-x_i^U) \vee (1-y_i^U)}},$$
$$\frac{\frac{(1-x_i^L) \wedge (1-y_i^L)}{(1-x_i^L) \vee (1-y_i^L)} + \frac{(1-x_i^U) \wedge (1-y_i^U)}{(1-x_i^U) \vee (1-y_i^U)}}{2}]. \quad (25)$$

$$INE^\alpha_{(g-mean,a-mean),(0,2)}(\widetilde{x},\widetilde{y}) = \frac{1}{3}\sum_{i=1}^{3}[\sqrt{\frac{2(x_i^L \wedge y_i^L)}{x_i^L + y_i^L} \cdot \frac{2(x_i^U \wedge y_i^U)}{x_i^U + y_i^U}}, \frac{\frac{2(x_i^L \wedge y_i^L)}{x_i^L + y_i^L} + \frac{2(x_i^U \wedge y_i^U)}{x_i^U + y_i^U}}{2}] \quad (26)$$

## 4. Similarity Measures for Interval Neutrosophic Sets

Theoretically, it is not difficult to construct the similarity between interval neutrosophic sets from the similarity between two interval neutrosophic values using some aggregation operators. In this section, we propose a method to construct the similarity measures between INSs using the interval neutrosophic fuzzy equivalencies.

**Definition 9** ([27]). *Let X be a finite set of objects. A function $\widetilde{S} : INS(X) \times INS(X) \rightarrow N_{[0,1]}$ is called a similarity measure for interval neutrosophic sets in X, if it satisfies the following properties:*

*($\widetilde{S}$1) $\widetilde{S}(A,B) = \widetilde{S}(B,A)$.*
*($\widetilde{S}$2) $\widetilde{S}(A,B) = [1,1]$, if and only if $A = B$.*
*($\widetilde{S}$3) $\widetilde{S}(\emptyset_{INS}, U_{INS}) = [0,0]$.*
*($\widetilde{S}$4) If $A \subseteq B \subseteq C$, then $\widetilde{S}(A,C) \leq \widetilde{S}(A,B), \widetilde{S}(A,C) \leq \widetilde{S}(B,C)$.*

**Theorem 3.** *Let $X = x_1, x_2, \cdots, x_n$ be a finite set of objects. Suppose that INE is an interval neutrosophic fuzzy equivalence on $\widetilde{D}^*$, then $\widetilde{S}_{INE} : INS(X) \times INS(X) \rightarrow N_{[0,1]}$ is a similarity measure, where for any $A, B \in INS(X)$,*

$$\widetilde{S}_{INE}(A,B) = \frac{1}{n}\sum_{i=1}^{n} INE(A(x_i), B(x_i)). \quad (27)$$

*If the weight vector $W = (w_1, w_2, \cdots, w_n)$ of objects X is added, $w_i \in [0,1]$ and $\sum_{i=1}^{n} w_i = 1$, then, the similarity measure of A and B is defined as follows:*

$$\widetilde{S}_{INE}(A,B) = \sum_{i=1}^{n} w_i \cdot INE(A(x_i), B(x_i)). \quad (28)$$

**Proof.** (1) For $\forall x_i \in X$, $INE(A(x_i), B(x_i)) = INE(B(x_i), A(x_i))$, then $\widetilde{S}_{INE}(A,B) = \widetilde{S}_{INE}(B,A)$.

(2) $\forall A, B \in INS(X)$, $\widetilde{S}_{INE}(A,B) = [1,1] \Leftrightarrow \forall x_i \in X, INE(A(x_i), B(x_i)) = [1,1] \Leftrightarrow \forall x_i \in X, A(x_i) = B(x_i) \Leftrightarrow A = B$.

(3) The conclusion immediately follows from Definitions 4 and 8.

(4) For any $A, B, C \in INS(X)$, if $A \subseteq B \subseteq C$, then $A(x_i) \le B(x_i) \le C(x_i)$, for any $x_i \in X$, thus $\widetilde{S}_{INE}(A, C) = \sum_{i=1}^{n} w_i \cdot INE(A(x_i), C(x_i)) \le \sum_{i=1}^{n} w_i \cdot INE(A(x_i), B(x_i)) = \widetilde{S}_{INE}(A, B)$. Similarly, we have $\widetilde{S}_{INE}(A, C) \le \widetilde{S}_{INE}(B, C)$. $\square$

According to the above theorem, using the fuzzy equivalencies $INE^{\alpha}_{(\mu,\varphi),(\theta,\varepsilon)}$ and $INE^{\beta}_{(\mu,\varphi),(\theta,\varepsilon)}$, it is easy to obtain the following similarity measures $\widetilde{S}$ for $INSs$. Suppose we take the weight $w_1 = w_2 = \cdots = w_n = \frac{1}{n}$, then

$$
\begin{aligned}
\widetilde{S}_{INE^{\alpha}_{(1,0),(0,1)}}(A, B) = \frac{1}{3n} \sum_{i=1}^{n} (&[\frac{T_A^L(x_i) \wedge T_B^L(x_i)}{T_A^L(x_i) \vee T_B^L(x_i)} \wedge \frac{T_A^U(x_i) \wedge T_B^U(x_i)}{T_A^U(x_i) \vee T_B^U(x_i)}, \\
&\frac{T_A^L(x_i) \wedge T_B^L(x_i)}{T_A^L(x_i) \vee T_B^L(x_i)} \vee \frac{T_A^U(x_i) \wedge T_B^U(x_i)}{T_A^U(x_i) \vee T_B^U(x_i)}] \\
+&[\frac{I_A^L(x_i) \wedge I_B^L(x_i)}{I_A^L(x_i) \vee I_B^L(x_i)} \wedge \frac{I_A^U(x_i) \wedge I_B^U(x_i)}{I_A^U(x_i) \vee I_B^U(x_i)}, \\
&\frac{I_A^L(x_i) \wedge I_B^L(x_i)}{I_A^L(x_i) \vee I_B^L(x_i)} \vee \frac{I_A^U(x_i) \wedge I_B^U(x_i)}{I_A^U(x_i) \vee I_B^U(x_i)}] \\
+&[\frac{F_A^L(x_i) \wedge F_B^L(x_i)}{F_A^L(x_i) \vee F_B^L(x_i)} \wedge \frac{F_A^U(x_i) \wedge F_B^U(x_i)}{F_A^U(x_i) \vee F_B^U(x_i)}, \\
&\frac{F_A^L(x_i) \wedge F_B^L(x_i)}{F_A^L(x_i) \vee F_B^L(x_i)} \vee \frac{F_A^U(x_i) \wedge F_B^U(x_i)}{F_A^U(x_i) \vee F_B^U(x_i)}])
\end{aligned}
\tag{29}
$$

$$
\begin{aligned}
\widetilde{S}_{INE^{\alpha}_{(1,0),(0,2)}}(A, B) = \frac{1}{3n} \sum_{i=1}^{n} (&[\frac{2(T_A^L(x_i) \wedge T_B^L(x_i))}{T_A^L(x_i) + T_B^L(x_i)} \wedge \frac{2(T_A^U(x_i) \wedge T_B^U(x_i))}{T_A^U(x_i) + T_B^U(x_i)}, \\
&\frac{2(T_A^L(x_i) \wedge T_B^L(x_i))}{T_A^L(x_i) + T_B^L(x_i)} \vee \frac{2(T_A^U(x_i) \wedge T_B^U(x_i))}{T_A^U(x_i) + T_B^U(x_i)}] \\
+&[\frac{2(I_A^L(x_i) \wedge I_B^L(x_i))}{I_A^L(x_i) + I_B^L(x_i)} \wedge \frac{2(I_A^U(x_i) \wedge I_B^U(x_i))}{I_A^U(x_i) + I_B^U(x_i)}, \\
&\frac{2(I_A^L(x_i) \wedge I_B^L(x_i))}{I_A^L(x_i) + I_B^L(x_i)} \vee \frac{2(I_A^U(x_i) \wedge I_B^U(x_i))}{I_A^U(x_i) + I_B^U(x_i)}] \\
+&[\frac{2(F_A^L(x_i) \wedge F_B^L(x_i))}{F_A^L(x_i) + F_B^L(x_i)} \wedge \frac{2(F_A^U(x_i) \wedge F_B^U(x_i))}{F_A^U(x_i) + F_B^U(x_i)}, \\
&\frac{2(F_A^L(x_i) \wedge F_B^L(x_i))}{F_A^L(x_i) + F_B^L(x_i)} \vee \frac{2(F_A^U(x_i) \wedge F_B^U(x_i))}{F_A^U(x_i) + F_B^U(x_i)}])
\end{aligned}
\tag{30}
$$

$$\widetilde{S}_{INE^{\alpha}_{(g-mean,a-mean),(0,1)}}(A,B) = \frac{1}{3n}\sum_{i=1}^{n}([\sqrt{\frac{T_A^L(x_i) \wedge T_B^L(x_i)}{T_A^L(x_i) \vee T_B^L(x_i)} \cdot \frac{T_A^U(x_i) \wedge T_B^U(x_i)}{T_A^U(x_i) \vee T_B^U(x_i)}},$$

$$\frac{\frac{T_A^L(x_i) \wedge T_B^L(x_i)}{T_A^L(x_i) \vee T_B^L(x_i)} + \frac{T_A^U(x_i) \wedge T_B^U(x_i)}{T_A^U(x_i) \vee T_B^U(x_i)}}{2}]$$

$$+[\sqrt{\frac{I_A^L(x_i) \wedge I_B^L(x_i)}{I_A^L(x_i) \vee I_B^L(x_i)} \cdot \frac{I_A^U(x_i) \wedge I_B^U(x_i)}{I_A^U(x_i) \vee I_B^U(x_i)}},$$

$$\frac{\frac{I_A^L(x_i) \wedge I_B^L(x_i)}{I_A^L(x_i) \vee I_B^L(x_i)} + \frac{I_A^U(x_i) \wedge I_B^U(x_i)}{I_A^U(x_i) \vee I_B^U(x_i)}}{2}] \qquad (31)$$

$$+[\sqrt{\frac{F_A^L(x_i) \wedge F_B^L(x_i)}{F_A^L(x_i) \vee F_B^L(x_i)} \cdot \frac{F_A^U(x_i) \wedge F_B^U(x_i)}{F_A^U(x_i) \vee F_B^U(x_i)}},$$

$$\frac{\frac{F_A^L(x_i) \wedge F_B^L(x_i)}{F_A^L(x_i) \vee F_B^L(x_i)} + \frac{F_A^U(x_i) \wedge F_B^U(x_i)}{F_A^U(x_i) \vee F_B^U(x_i)}}{2}])$$

$$\widetilde{S}_{INE^{\alpha}_{(g-mean,a-mean),(0,2)}}(A,B) = \frac{1}{3n}\sum_{i=1}^{n}([\sqrt{\frac{2(T_A^L(x_i) \wedge T_B^L(x_i))}{T_A^L(x_i) + T_B^L(x_i)} \cdot \frac{2(T_A^U(x_i) \wedge T_B^U(x_i))}{T_A^U(x_i) + T_B^U(x_i)}},$$

$$\frac{\frac{2(T_A^L(x_i) \wedge T_B^L(x_i))}{T_A^L(x_i) + T_B^L(x_i)} + \frac{2(T_A^U(x_i) \wedge T_B^U(x_i))}{T_A^U(x_i) + T_B^U(x_i)}}{2}]$$

$$+[\sqrt{\frac{2(I_A^L(x_i) \wedge I_B^L(x_i))}{I_A^L(x_i) + I_B^L(x_i)} \cdot \frac{2(I_A^U(x_i) \wedge I_B^U(x_i))}{I_A^U(x_i) + I_B^U(x_i)}},$$

$$\frac{\frac{2(I_A^L(x_i) \wedge I_B^L(x_i))}{I_A^L(x_i) + I_B^L(x_i)} + \frac{2(I_A^U(x_i) \wedge I_B^U(x_i))}{I_A^U(x_i) + I_B^U(x_i)}}{2}] \qquad (32)$$

$$+[\sqrt{\frac{2(F_A^L(x_i) \wedge F_B^L(x_i))}{F_A^L(x_i) + F_B^L(x_i)} \cdot \frac{2(F_A^U(x_i) \wedge F_B^U(x_i))}{F_A^U(x_i) + F_B^U(x_i)}},$$

$$\frac{\frac{2(F_A^L(x_i) \wedge F_B^L(x_i))}{F_A^L(x_i) + F_B^L(x_i)} + \frac{2(F_A^U(x_i) \wedge F_B^U(x_i))}{F_A^U(x_i) + F_B^U(x_i)}}{2}])$$

$$\widetilde{S}_{INE^{\beta}_{(g-mean,a-mean),(0,1)}}(A,B) =$$

$$\frac{1}{3n}\sum_{i=1}^{n}\left(\left[\sqrt{\frac{(1-T_A^L(x_i))\wedge(1-T_B^L(x_i))}{(1-T_A^L(x_i))\vee(1-T_B^L(x_i))}\cdot\frac{(1-T_A^U(x_i))\wedge(1-T_B^U(x_i))}{(1-T_A^U(x_i))\vee(1-T_B^U(x_i))}},\right.\right.$$

$$\frac{\frac{(1-T_A^L(x_i))\wedge(1-T_B^L(x_i))}{(1-T_A^L(x_i))\vee(1-T_B^L(x_i))}+\frac{(1-T_A^U(x_i))\wedge(1-T_B^U(x_i))}{(1-T_A^U(x_i))\vee(1-T_B^U(x_i))}}{2}\right]$$

$$+\left[\sqrt{\frac{(1-I_A^L(x_i))\wedge(1-I_B^L(x_i))}{(1-I_A^L(x_i))\vee(1-I_B^L(x_i))}\cdot\frac{(1-I_A^U(x_i))\wedge(1-I_B^U(x_i))}{(1-I_A^U(x_i))\vee(1-I_B^U(x_i))}},\right. \tag{33}$$

$$\frac{\frac{(1-I_A^L(x_i))\wedge(1-I_B^L(x_i))}{(1-I_A^L(x_i))\vee(1-I_B^L(x_i))}+\frac{(1-I_A^U(x_i))\wedge(1-I_B^U(x_i))}{(1-I_A^U(x_i))\vee(1-I_B^U(x_i))}}{2}\right]$$

$$+\left[\sqrt{\frac{(1-F_A^L(x_i))\wedge(1-F_B^L(x_i))}{(1-F_A^L(x_i))\vee(1-F_B^L(x_i))}\cdot\frac{(1-F_A^U(x_i))\wedge(1-F_B^U(x_i))}{(1-F_A^U(x_i))\vee(1-F_B^U(x_i))}},\right.$$

$$\left.\left.\frac{\frac{(1-F_A^L(x_i))\wedge(1-F_B^L(x_i))}{(1-F_A^L(x_i))\vee(1-F_B^L(x_i))}+\frac{(1-F_A^U(x_i))\wedge(1-F_B^U(x_i))}{(1-F_A^U(x_i))\vee(1-F_B^U(x_i))}}{2}\right]\right).$$

## 5. Decision Applications

Assume that $A=\{A_i|1\le i\le m\}$ is the set of alternatives and that $C=\{C_j|1\le j\le n\}$ is a collection of attributes. For decision making, it is required to provide the information $\widetilde{\rho}^{ij}$ which is the evaluation on the alternative $A_i$ for the attribute $C_j$. The evaluation information can be represented as an interval neutrosophic value $\widetilde{\rho}^{ij}=(T_{\widetilde{\rho}^{ij}},I_{\widetilde{\rho}^{ij}},F_{\widetilde{\rho}^{ij}})=([T_{\widetilde{\rho}^{ij}}^L,T_{\widetilde{\rho}^{ij}}^U],[I_{\widetilde{\rho}^{ij}}^L,I_{\widetilde{\rho}^{ij}}^U],[F_{\widetilde{\rho}^{ij}}^L,F_{\widetilde{\rho}^{ij}}^U])$ [21,29,33]. When all the evaluations of the alternatives are provided, the interval neutrosophic decision matrix $(\widetilde{\rho}^{ij})_{m\times n}$ can be constructed. Assume that the weight vector of attributes $W=(w_1,w_2,\cdots,w_n)$ is given by the experts, where $0\le w_i\le 1(1\le j\le n)$ and $\sum_{i=1}^{n}w_i=1$.

In order to obtain the optimal alternatives, the computational procedure can be summarized as:

- Step 1: Based on the decision matrix $(\widetilde{\rho}^{ij})_{m\times n}$, we can compute the similarity $INE^{\alpha}_{(\mu,\varphi),(\theta,\varepsilon)}(\widetilde{\rho}^{ij},\widetilde{\rho}^*)$ by using the equations on Theorem 2, where $\widetilde{\rho}^*$ is denoted as the positive ideal interval neutrosophic value in $\widetilde{D}^*$ with respect to the product order relation $\le$.

- Step 2: We can aggregate these similarities $INE^{\alpha}_{(\mu,\varphi),(\theta,\varepsilon)}(\widetilde{\rho}^{ij},\widetilde{\rho}^*)$ to obtain the similarity degree $\widetilde{S}(A_i,\widetilde{\rho}^*)$ by

$$\widetilde{S}(A_i,\widetilde{\rho}^*)=w_1\cdot INE^{\alpha}_{(\mu,\varphi),(\theta,\varepsilon)}(\widetilde{\rho}^{i1},\widetilde{\rho}^*)+w_2\cdot INE^{\alpha}_{(\mu,\varphi),(\theta,\varepsilon)}(\widetilde{\rho}^{i2},\widetilde{\rho}^*)$$
$$+\cdots+w_n\cdot INE^{\alpha}_{(\mu,\varphi),(\theta,\varepsilon)}(\widetilde{\rho}^{in},\widetilde{\rho}^*). \tag{34}$$

- Step 3: By ranking these similarity degrees $\widetilde{S}(A_i,\widetilde{\rho}^*)$, we can select the best alternative $A_k$, in which the best alternative $A_k$ satisfies the condition $\widetilde{S}(A_k,\widetilde{\rho}^*)=\max\{\widetilde{S}(A_i,\widetilde{\rho}^*)|1\le i\le m\}$.

### 5.1. Decision Applications in Resource Offloading of Edge Computing

An offloading resource is considered a key segment for edge computing. In order to ensure the normal operation of services for clients, redundant work is allocated to edge servers based on the load capacity. Therefore, selecting and matching clients and edge servers is a very important decision problem for resource offloading. In this paper, we

intend to present a multi-attribute decision-making method based on interval neutrosophic valued fuzzy equivalencies.

**Example 3.** *As shown in Figure 1, we set n clients and m edge servers. With the goal of achieving a reasonable usage of the computing resource, when the load of the client is high, resources must be offloaded to the edge server. That is, when the computing usage of the client end quickly exceeds its own responsiveness, it is necessary to reasonably allocate the excess computing amount to the server and reduce the burden. This can enable a faster response and improve the robustness of the network, thus greatly improving the computing performance. As shown in Tables 1 and 2, we use four clients and three edge servers as examples. These all have three criteria, including CPU, memory, and network. $C_1$ is the quantification of CPU. $C_2$ is memory. $C_3$ is network. Specifically, Figure 2 depicts how we quantify three criteria for clients, in practice, the reference interval set to $[0.5, 0.9]$ for CPU, $[0.4, 0.8]$ for memory, $[0.6, 0.9]$ for network. Figure 3 shows the quantification for edge servers, the reference interval set to $[0.3, 0.8]$ for CPU, $[0.4, 0.8]$ for memory, $[0.2, 0.6]$ for network. Due to the rates of three criteria for clients being higher, this means that the client needs to perform an offloading task. Furthermore, experiential values measure the intervals of T, I, and F for three criteria. A denotes the client and B denotes the edge server. Using the interval-valued intuitionistic fuzzy information, the idle $\widetilde{S}(B_i, \widetilde{\rho}^*)$ can be selected to respond to the high-occupancy $\widetilde{S}(A_j, \widetilde{\rho}^*)$.*

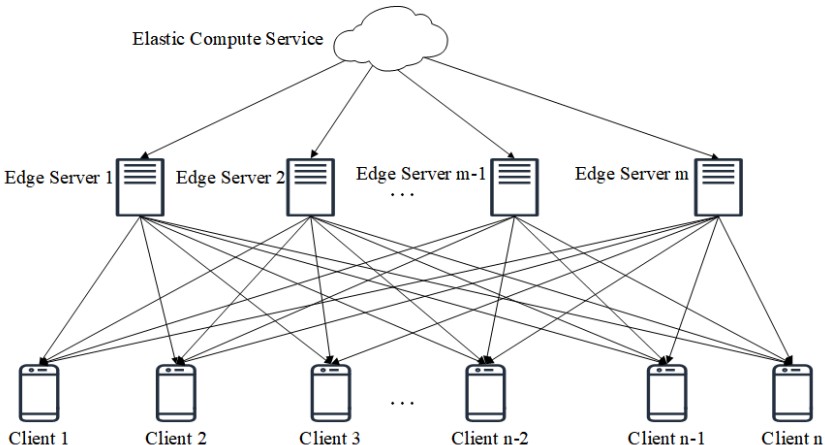

**Figure 1.** The framework of edge computing.

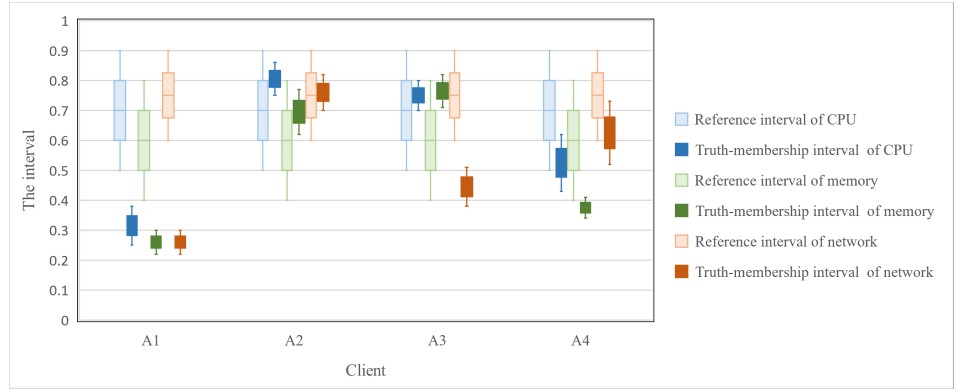

**Figure 2.** The visualization of the truth-membership interval of clients compared with the reference interval of three criteria.

**Table 1.** The criteria of clients in Example 3.

|       | $C_1$ | $C_2$ | $C_3$ |
|-------|-------|-------|-------|
| $A_1$ | $([0.25, 0.38], [0.10, 0.20], [0.55, 0.60])$ | $([0.22, 0.30], [0.11, 0.21], [0.51, 0.67])$ | $([0.22, 0.30], [0.12, 0.20], [0.55, 0.70])$ |
| $A_2$ | $([0.75, 0.86], [0.11, 0.13], [0.12, 0.13])$ | $([0.62, 0.77], [0.16, 0.22], [0.16, 0.23])$ | $([0.70, 0.82], [0.11, 0.20], [0.10, 0.18])$ |
| $A_3$ | $([0.70, 0.80], [0.17, 0.20], [0.10, 0.20])$ | $([0.71, 0.82], [0.13, 0.15], [0.13, 0.15])$ | $([0.38, 0.51], [0.44, 0.50], [0.12, 0.45])$ |
| $A_4$ | $([0.43, 0.62], [0.23, 0.30], [0.24, 0.31])$ | $([0.34, 0.41], [0.16, 0.30], [0.30, 0.50])$ | $([0.52, 0.73], [0.20, 0.30], [0.18, 0.26])$ |

**Table 2.** The criteria of edge servers in Example 3.

|       | $C_1$ | $C_2$ | $C_3$ |
|-------|-------|-------|-------|
| $B_1$ | $([0.23, 0.30], [0.22, 0.35], [0.42, 0.70])$ | $([0.23, 0.34], [0.22, 0.30], [0.47, 0.66])$ | $([0.40, 0.52], [0.31, 0.40], [0.20, 0.48])$ |
| $B_2$ | $([0.34, 0.53], [0.34, 0.50], [0.16, 0.47])$ | $([0.47, 0.79], [0.37, 0.40], [0.13, 0.21])$ | $([0.15, 0.27], [0.22, 0.31], [0.54, 0.70])$ |
| $B_3$ | $([0.13, 0.22], [0.17, 0.20], [0.67, 0.78])$ | $([0.31, 0.40], [0.26, 0.38], [0.31, 0.60])$ | $([0.10, 0.20], [0.15, 0.20], [0.70, 0.80])$ |

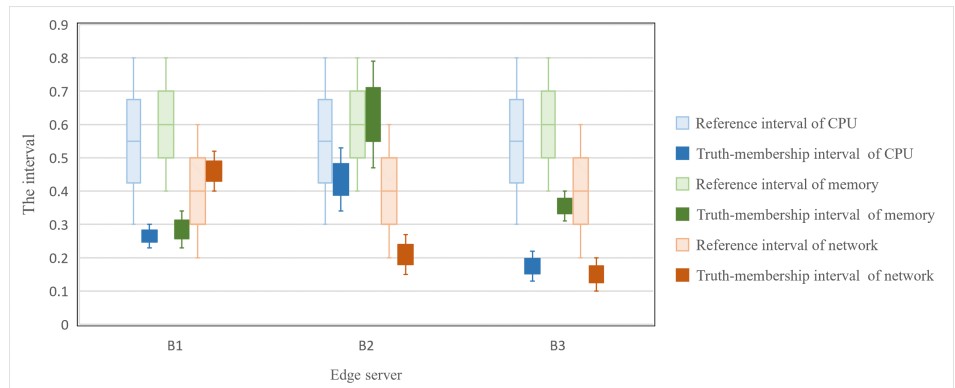

**Figure 3.** The visualization of the truth-membership interval of edge servers compared with the reference interval of three criteria.

*Suppose that the weights of $C_1$, $C_2$, and $C_3$ are 0.4, 0.35, and 0.25, respectively. Then, the most desirable alternative for clients is as follows.*

*Step 1: By the equation of Theorem 2, $\widetilde{\rho}_1^* = ([0.8, 0.9], [0.1, 0.2], [0.1, 0.2])$, we have*

$$INE_{(1,0),(0,1)}^{\alpha}(\widetilde{\rho}_1^{11}, \widetilde{\rho}_1^*) = \frac{1}{3}([\frac{0.25}{0.8} \wedge \frac{0.38}{0.9}, \frac{0.25}{0.8} \vee \frac{0.38}{0.9}] + [\frac{0.10}{0.10} \wedge \frac{0.20}{0.20}, \frac{0.10}{0.10} \vee \frac{0.20}{0.20}]$$
$$+ [\frac{0.10}{0.55} \wedge \frac{0.20}{0.60}, \frac{0.10}{0.55} \vee \frac{0.20}{0.60}]) = [0.4981, 0.5852].$$

*For the same reason, we have the following results,*
$INE_{(1,0),(0,1)}^{\alpha}(\widetilde{\rho}_1^{12}, \widetilde{\rho}_1^*) = [0.46, 0.5281]$, $INE_{(1,0),(0,1)}^{\alpha}(\widetilde{\rho}_1^{13}, \widetilde{\rho}_1^*) = [0.4301, 0.5397]$,
$INE_{(1,0),(0,1)}^{\alpha}(\widetilde{\rho}_1^{21}, \widetilde{\rho}_1^*) = [0.7458, 0.8993]$, $INE_{(1,0),(0,1)}^{\alpha}(\widetilde{\rho}_1^{22}, \widetilde{\rho}_1^*) = [0.675, 0.8781]$,
$INE_{(1,0),(0,1)}^{\alpha}(\widetilde{\rho}_1^{23}, \widetilde{\rho}_1^*) = [0.8946, 0.9704]$, $INE_{(1,0),(0,1)}^{\alpha}(\widetilde{\rho}_1^{31}, \widetilde{\rho}_1^*) = [0.8211, 0.9629]$,
$INE_{(1,0),(0,1)}^{\alpha}(\widetilde{\rho}_1^{32}, \widetilde{\rho}_1^*) = [0.7958, 0.8165]$, $INE_{(1,0),(0,1)}^{\alpha}(\widetilde{\rho}_1^{33}, \widetilde{\rho}_1^*) = [0.3822, 0.6]$,
$INE_{(1,0),(0,1)}^{\alpha}(\widetilde{\rho}_1^{41}, \widetilde{\rho}_1^*) = [0.4630, 0.6669]$, $INE_{(1,0),(0,1)}^{\alpha}(\widetilde{\rho}_1^{42}, \widetilde{\rho}_1^*) = [0.4611, 0.5074]$,
$INE_{(1,0),(0,1)}^{\alpha}(\widetilde{\rho}_1^{43}, \widetilde{\rho}_1^*) = [0.5685, 0.7490]$.

*Step 2: By Equation (34), we have*

$$\widetilde{S}(A_1, \widetilde{\rho}_1^*) = 0.4 \times [0.4981, 0.5852] + 0.35 \times [0.46, 0.5281] + 0.25 \times [0.4301, 0.5397] = [0.4678, 0.5538].$$

$$\widetilde{S}(A_2, \widetilde{\rho}_1^*) = [0.7582, 0.9097], \widetilde{S}(A_3, \widetilde{\rho}_1^*) = [0.7025, 0.8209], \widetilde{S}(A_4, \widetilde{\rho}_1^*) = [0.4887, 0.6316].$$

*Step 3: According to the product order $\leq$ of the interval-values, we can rank these similarity measures:*

$$\widetilde{S}(A_2,\widetilde{\rho}_1^*) > \widetilde{S}(A_3,\widetilde{\rho}_1^*) > \widetilde{S}(A_4,\widetilde{\rho}_1^*) > \widetilde{S}(A_1,\widetilde{\rho}_1^*)$$

*The most desirable alternative for edge servers is as follows.*

*Step 1: By the equation of Theorem 2, $\widetilde{\rho}_2^* = ([0.1,0.2],[0.8,0.9],[0.8,0.9])$, we have*

$$INE_{(1,0),(0,1)}^{\alpha}(\widetilde{\rho}_2^{11},\widetilde{\rho}_2^*) = \frac{1}{3}([\frac{0.10}{0.23} \wedge \frac{0.20}{0.30}, \frac{0.10}{0.23} \vee \frac{0.20}{0.30}] + [\frac{0.22}{0.80} \wedge \frac{0.35}{0.90}, \frac{0.22}{0.80} \vee \frac{0.35}{0.90}]$$
$$+ [\frac{0.42}{0.80} \wedge \frac{0.70}{0.90}, \frac{0.42}{0.80} \vee \frac{0.70}{0.90}]) = [0.4116, 0.6111].$$

*For the same reason, we have the following results,*
$INE_{(1,0),(0,1)}^{\alpha}(\widetilde{\rho}_2^{12},\widetilde{\rho}_2^*) = [0.4324, 0.5516]$, $INE_{(1,0),(0,1)}^{\alpha}(\widetilde{\rho}_2^{13},\widetilde{\rho}_2^*) = [0.2958, 0.4541]$,
$INE_{(1,0),(0,1)}^{\alpha}(\widetilde{\rho}_2^{21},\widetilde{\rho}_2^*) = [0.3064, 0.4850]$, $INE_{(1,0),(0,1)}^{\alpha}(\widetilde{\rho}_2^{22},\widetilde{\rho}_2^*) = [0.2732, 0.3163]$,
$INE_{(1,0),(0,1)}^{\alpha}(\widetilde{\rho}_2^{23},\widetilde{\rho}_2^*) = [0.5389, 0.6210]$, $INE_{(1,0),(0,1)}^{\alpha}(\widetilde{\rho}_2^{31},\widetilde{\rho}_2^*) = [0.6064, 0.6659]$,
$INE_{(1,0),(0,1)}^{\alpha}(\widetilde{\rho}_2^{32},\widetilde{\rho}_2^*) = [0.3450, 0.5296]$, $INE_{(1,0),(0,1)}^{\alpha}(\widetilde{\rho}_2^{33},\widetilde{\rho}_2^*) = [0.6875, 0.7037]$.

*Step 2: By Equation (34), we have*

$$\widetilde{S}(B1,\widetilde{\rho}_2^*) = 0.4 \times [0.4116, 0.6111] + 0.35 \times [0.4324, 0.5516]$$
$$+ 0.25 \times [0.2958, 0.4541] = [0.3899, 0.5510].$$

$\widetilde{S}(B_2,\widetilde{\rho}_2^*) = [0.3529, 0.46]$, $\widetilde{S}(B_3,\widetilde{\rho}_2^*) = [0.5352, 0.6276]$.

*Step 3: According to the product order $\leq$ of the interval-values, we can rank these similarity measures:*

$$\widetilde{S}(B_3,\widetilde{\rho}_2^*) > \widetilde{S}(B_1,\widetilde{\rho}_2^*) > \widetilde{S}(B_2,\widetilde{\rho}_2^*)$$

*As shown in Figure 4, it depicts the compared results of clients and edge servers. For clients, $A_2$ is the highest occupancy and needs to offload resources to the edge server. According to the compared results between the edge servers, $B_3$ with less resource usage is the most suitable for receiving services from the client $A_2$.*

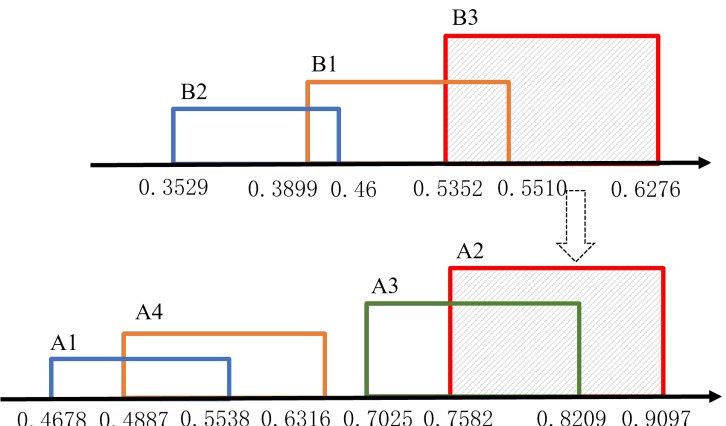

**Figure 4.** The visualization of the matching result.

### 5.2. Comparative Analysis of Decision Application

Let us consider the decision-making problem adapted from [21,22,30]. Ye [21] presented the multi-attributes decision-making method using the Hamming and Euclidean distance. Sahin [22] proposed two multi-criteria decision-making methods using the interval neutrosophic cross-entropy between an alternative and the ideal alternative. Yang [30]

presented the multi-attributes decision-making method based on the similarity measure using a new inclusion relationship.

**Example 4.** *Suppose that there is a panel with four possible alternatives to invest the money: (1) $A_1$ is a food company; (2) $A_2$ is a car company; (3) $A_3$ is an arms company; and (4) $A_4$ is a computer company. The investment company must make a decision according to the three criteria given below: (1) $C_1$ is the growth analysis; (2) $C_2$ is the risk analysis; and (3) $C_3$ is the environmental impact analysis. Using the interval-valued intuitionistic fuzzy information, the decision maker evaluates the four possible alternatives under the above three criteria and the evaluation are expressed as three interval neutrosophic sets (Table 3).*

**Table 3.** The evaluation of alternatives.

|  | $C_1$ | $C_2$ | $C_3$ |
|---|---|---|---|
| $A_1$ | $([0.4, 0.5], [0.2, 0.3], [0.3, 0.4])$ | $([0.4, 0.6], [0.1, 0.3], [0.2, 0.4])$ | $([0.7, 0.9], [0.2, 0.3], [0.4, 0.5])$ |
| $A_2$ | $([0.6, 0.7], [0.1, 0.2], [0.2, 0.3])$ | $([0.6, 0.7], [0.1, 0.2], [0.2, 0.3])$ | $([0.3, 0.6], [0.3, 0.5], [0.8, 0.9])$ |
| $A_3$ | $([0.3, 0.6], [0.2, 0.3], [0.3, 0.4])$ | $([0.5, 0.6], [0.2, 0.3], [0.3, 0.4])$ | $([0.4, 0.5], [0.2, 0.4], [0.7, 0.9])$ |
| $A_4$ | $([0.7, 0.8], [0.0, 0.1], [0.1, 0.2])$ | $([0.6, 0.7], [0.1, 0.2], [0.1, 0.3])$ | $([0.6, 0.7], [0.3, 0.4], [0.8, 0.9])$ |

*Suppose that the weights of $C_1$, $C_2$, and $C_3$ are 0.35, 0.25, and 0.4, respectively. Then, we use the approach proposed to obtain the most desirable alternative.*

*Step 1: The positive ideal interval neutrosophic value is $\widetilde{\rho}^* = ([1,1], [0,0], [0,0])$; by the equation of Theorem 2, we have*

$$INE^\alpha_{(1,0),(0,1)}(\widetilde{\rho}^{11}, \widetilde{\rho}^*) = \frac{1}{3}([\frac{0.4}{1} \wedge \frac{0.5}{1}, \frac{0.4}{1} \vee \frac{0.5}{1}] + [\frac{0}{0.2} \wedge \frac{0}{0.3}, \frac{0}{0.2} \vee \frac{0}{0.3}]$$
$$+ [\frac{0}{0.3} \wedge \frac{0}{0.4}, \frac{0}{0.3} \vee \frac{0}{0.4}]) = [\frac{0.4}{3}, \frac{0.5}{3}].$$

*For the same reason, we have the following results:*
$INE^\alpha_{(1,0),(0,1)}(\widetilde{\rho}^{12}, \widetilde{\rho}^*) = [\frac{0.4}{3}, 0.2]$, $INE^\alpha_{(1,0),(0,1)}(\widetilde{\rho}^{13}, \widetilde{\rho}^*) = [\frac{0.7}{3}, 0.3]$,

$INE^\alpha_{(1,0),(0,1)}(\widetilde{\rho}^{21}, \widetilde{\rho}^*) = [0.2, \frac{0.7}{3}]$, $INE^\alpha_{(1,0),(0,1)}(\widetilde{\rho}^{22}, \widetilde{\rho}^*) = [0.2, \frac{0.7}{3}]$,

$INE^\alpha_{(1,0),(0,1)}(\widetilde{\rho}^{23}, \widetilde{\rho}^*) = [0.1, 0.2]$, $INE^\alpha_{(1,0),(0,1)}(\widetilde{\rho}^{31}, \widetilde{\rho}^*) = [0.1, 0.2]$,

$INE^\alpha_{(1,0),(0,1)}(\widetilde{\rho}^{32}, \widetilde{\rho}^*) = [\frac{0.5}{3}, 0.2]$, $INE^\alpha_{(1,0),(0,1)}(\widetilde{\rho}^{33}, \widetilde{\rho}^*) = [\frac{0.4}{3}, \frac{0.5}{3}]$,

$INE^\alpha_{(1,0),(0,1)}(\widetilde{\rho}^{41}, \widetilde{\rho}^*) = [\frac{0.7}{3}, \frac{0.8}{3}]$, $INE^\alpha_{(1,0),(0,1)}(\widetilde{\rho}^{42}, \widetilde{\rho}^*) = [0.2, \frac{0.7}{3}]$,

$INE^\alpha_{(1,0),(0,1)}(\widetilde{\rho}^{43}, \widetilde{\rho}^*) = [0.2, \frac{0.7}{3}]$.

*Step 2: By Equation (34), we have*

$$\widetilde{S}(A_1, \widetilde{\rho}^*) = 0.35 \times [\frac{0.4}{3}, \frac{0.5}{3}] + 0.25 \times [\frac{0.4}{3}, 0.2] + 0.4 \times [\frac{0.7}{3}, 0.3] = [0.1733, 0.2283].$$

*Similarly, $\widetilde{S}(A_2, \widetilde{\rho}^*) = [0.16, 0.22]$, $\widetilde{S}(A_3, \widetilde{\rho}^*) = [0.13, 0.1867]$, $\widetilde{S}(A_4, \widetilde{\rho}^*) = [0.2117, 0.245]$.*

*Step 3: According to the product order $\leq$ of the interval-values, we can rank these similarity measures:*
$$\widetilde{S}(A_4, \widetilde{\rho}^*) > \widetilde{S}(A_1, \widetilde{\rho}^*) > \widetilde{S}(A_2, \widetilde{\rho}^*) > \widetilde{S}(A_3, \widetilde{\rho}^*)$$

*Therefore, the most desirable alternative is $A_4$.*

In order to validate the feasibility of the proposed decision-making method, a comparative study was conducted with other methods as shown in Table 4.

**Table 4.** The comparison of results of the decision-making methods in Example 4.

| Decision-Making Methods | The Order of the Alternatives |
|---|---|
| Ye's method based on the Hamming distance | $A_4 > A_2 > A_3 > A_1$ |
| Ye's method based on the Euclidean distance | $A_2 > A_4 > A_3 > A_1$ |
| Sahin's method based on cross-entropy | $A_4 > A_1 > A_2 > A_3$ |
| Yang's method based on the new inclusion relationship | $A_4 > A_1 > A_2 > A_3$ |
| The method based on interval neutrosophic fuzzy equivalencies | $A_4 > A_1 > A_2 > A_3$ |

For this example, using the new similarity measure based on the interval neutrosophic fuzzy equivalencies proposed in this paper, we obtain the same ranking order of alternatives as in [22,30]. This shows that the similarity measure of interval neutrosophic sets proposed in this paper are effective and efficient.

## 6. Conclusions

The study of information measures is an important research topic in uncertain information processing. This paper presents a new approach to constructing similarity measures for interval neutrosophic sets using fuzzy equivalencies. The proposed method is based on the framework of interval neutrosophic sets and is designed to retain more interval-valued information. The similarity degree is expressed as an interval, and different similarities can be obtained by selecting parameters in the fuzzy equivalence, depending on practical situations. The effectiveness of the proposed method is discussed in edge computing applications, where it can be used to select and match clients and edge servers for resource offloading. An illustrative example is provided to verify the proposed method's ability to find a reasonable client and edge server.

In further research, on the one hand, since it is closely related to the parameters $\theta, \varepsilon$ and the aggregation manner $\mu, \varphi$ for the definition of the interval neutrosophic valued fuzzy equivalency $INE^{\alpha}_{(\mu,\varphi),(\theta,\varepsilon)}$, if there are different values for these parameters, we will subsequently discuss the relationship of these similarity measures. On the other hand, there is another parallel generalization of fuzzy sets known by T-spherical fuzzy sets (T-SFSs) presented by [4,5], which is characterized by the degrees of membership $s$, abstinence $i$, and non-membership $d$ of the form $(s, i, d)$, and satisfies $0 \leq s^n + i^n + d^n \leq 1$ for the positive integer $n$. This is similar to the triple structure of NSs in form, but we think that each element in the triplet $(T, I, F)$ of NSs is independent, as long as the triplet satisfies $0 \leq \sup T + \sup I + \sup F \leq 3$, so both NSs and T-SFSs enlarge the discussion space of fuzzy information, and meanwhile, each has its own advantages. The comparison and connection between NSs and T-SFSs will also be directions of future research works.

**Author Contributions:** Conceptualization, Q.L.; methodology, Q.L.; validation, Q.L. and X.W.; formal analysis, Q.L.; investigation, Q.L.; resources, Q.L. and M.K.; data curation, Q.L. and X.W.; writing—original draft preparation, Q.L.; writing—review and editing, Q.L.; visualization, Q.L. and X.W.; supervision, M.K. and K.Q.; funding acquisition, Q.L. All authors have read and agreed to the published version of the manuscript.

**Funding:** This research is supported by the National Key Research and Development Program of China (2020AAA0107702), Sichuan Provincial Administration of Traditional Chinese Medicine Special Project on Traditional Chinese Medicine Research (No.2018QN068), and the key R&D project jointly implemented by Sichuan and Chongqing in 2020 (cstc2020jscx-cylhX0004).

**Data Availability Statement:** Not applicable.

**Conflicts of Interest:** The authors declare no conflict of interest.

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
