# Peer review of "Similarity Measure for Interval Neutrosophic Sets and Its Decision Application in Resource Offloading of Edge Computing"

_electronics, doi:10.3390/electronics12081931_

Round 1

Reviewer 1 Report

Dear authors,

please find attached pdf.

Author Response

Concern 1: References are considerably few and not up-to-date. I would highly suggest you increase the number of references (add 10-15 references more).

Author action: Thank you for your comment. We have updated our references by adding more relevant ones and colored red in the revised manuscript.

Concern 2: Line 190 (add space): For anyx,y For any x, y.

Author action: Thank you for your comment, we have updated the writing in line 202 and colored red.

Concern 3: Line 229 (remove :) : and it is defined as follows: for any a, b and it is defined as follows for any a, b

Author action: Thank you for your comment, we have removed : and added , in line 241 with colored red to make it easier to understand.

Concern 4: Equations are hard to follow. When you present examples such as: ”if µ = 0, ϕ = 1, θ = 0, ϵ = 0” you can call the relevant Equation where you try different hyperparameters. For example, you could say, On Eq. 14, if we have these hyperparameters then we have Eq. 15, and so on.

Author action: Thank you for your comment, we have added the relevant equations in line 263-270, and line 274 , line 277 for easier learning our method and colored red.

Concern 5: Line 251: for 0 λ1 λ2 1 suppose that . . . change to For 0 λ1 ≤λ2 1 Suppose that. So capitalise For and Suppose.

Author action: Thank you for your comment, we have updated the wrong writing in Line 263 colored red.

Concern 6: Do the same for 258 as you start text after full-stop on Eq. 19.

Author action: Thank you for your comment, we have updated the wrong writing in Line 272 colored red.

Concern 7: Line 282: (1)Suppose (1) Suppose. Add one space.

Author action: Thank you for your comment, we have updated the wrong writing in Line 297 colored red.

Concern 8: Same for Lines 285, 288 and 294.

Author action: Thank you for your comment, we have updated the wrong writing in Line 300, 303 and 309 colored red.

Concern 9: Eq. 42, please try to fit it in-page. It goes outside the area of the manuscript.

Author action: Thank you for your comment, we have adjusted the format of Eq. (33), which is Eq.42 before.

Concern 10: You mention fuzzy sets and similar works. However, one related work using Neuro-fuzzy systems such as https://doi.org/10.3390/a16030151 can be taken into account for adding in the introduction or for comparison.

Author action: Thank you for your comment, we have added the relevant work for comparison in Introduction.

Concern 11: Lastly, kindly increase the conclusions and future work. I would recommend extending it to a single page if possible.

Author action: Thank you for your comment, we have updated the conclusion and future work to be clear in a single page.

Concern 12: Almost, all the introduction section need to be re-phrased and re-write from the beginning, Lines 28-86, 92-105, 112-114, and 116-118.

Author action: Thank you for your comment, we have updated our Introduction to improve the readability and make it easier to understand our work.

Concern 13: Definition 4, Lines 168-181. Kindly, add the reference of your work. http://fs.unm.edu/neut/RelationshipsAmongSseveral.pdf

Author action: Thank you for your comment, we have added the reference in Definition 4, Lines 180-184.

Concern 14: Definition 9, Lines 315-336. Kindly, add the reference of your work.

Author action: Thank you for your comment, we have added the reference in line 327 and provide the construction method about similarity.

Concern 15: Section 5, Lines 337-339. These need to be rewritten.

Author action: Thank you for your comment, we have update the writing in Lines 351-375 to make it clear know our work and goal.

Concern 16: Section 5, Lines 349-352. Kindly, add in the assumption the reference of https://doi.org/10.1007/s00500-020-04930-8

Author action: Thank you for your comment, we have added the reference in Line 355.

Concern 17: Section: Conclusions, Lines 628-639.

Author action: Thank you for your comment, we have updated our conclusion in Lines 469-478 colored red in the revised manuscript.

Reviewer 2 Report

In this paper, some similarity measures are introduced for the interval valued neutrosophic sets based on the fuzzy equivalence classes and then applied to the decision-making problem. The article is formalized well and is interesting. Some revisions are required as follows

1.       Discuss the novelty of the interval valued neutrosophic sets in abstracts.

2.        Elaborate the references 14-16.

3.       Give a solid motivation in the introduction of your article.

4.       What about similarity measure without weights of attributes? Please provide the definition of the similarity measure on interval valued neutrosophic set in start of section 4 of the section 4.

5.       Adjust the Eqn. 42.

6.       Compare your obtained results with existing alternative approaches.

7.       There is another parallel generazation of fuzzy sets known by T-spherical fuzzy sets. An approach toward decision-making and medical diagnosis problems using the concept of spherical fuzzy sets. Please compare it with neutorosiphic set thoeretically.

8.       The introduction can be improved with updated literature i.e., Performance evaluation of solar energy cells using the interval-valued T-spherical fuzzy Bonferroni mean operators

9. The similarity is 38%, please reduce it.

Author Response

Concern 1: Discuss the novelty of the interval valued neutrosophic sets in abstracts.

Author action: Thank you for your comment, we have updated the abstract to stress the novelty about the proposed method and colored red in the revised manuscript in Line 1-5.

Concern 2: Elaborate the references 14-16.

Author action: Thank you for your comment, we have updated the references and colored red in the revised manuscript.

Concern 3: Give a solid motivation in the introduction of your article.

Author action: Thank you for your comment, we have updated our Introduction to improve the readability and make it easier to understand our work

Concern 4: What about similarity measure without weights of attributes? Please provide the definition of the similarity measure on interval valued neutrosophic set in start of section 4 of the section 4

Author action: Thank you for your comment, we have added the similarity measure equations without weights in Section 4, theorem 3 colored red.

Concern 5: Adjust the Eqn. 42.

Author action: Thank you for your comment, we have adjusted the format of Eq. (33), which is Eq.42 before.

Concern 6: Compare your obtained results with existing alternative approaches.

Author action: Thank you for your comment, we have compared the obtained results and details in Section 5.2 colored red.

Concern 7: There is another parallel generazation of fuzzy sets known by T-spherical fuzzy sets. An approach toward decision-making and medical diagnosis problems using the concept of spherical fuzzy sets. Please compare it with neutorosiphic set thoeretically.

Author action: Thank you for your comment, we have added the compared results in Section 5.2. Meanwhile, we also discussed the relevant work in the Section 6, future work.

Concern 8: The introduction can be improved with updated literature i.e., Performance evaluation of solar energy cells using the interval-valued T-spherical fuzzy Bonferroni mean operators

Author action: Thank you for your comment, we have added the relevant work for comparison in Introduction.

Concern 9: The similarity is 38%, please reduce it.

Author action: Thank you for your comment, we have updated the manuscript to highlight our idea and work with low similarity.

Round 2

Reviewer 1 Report

Dear authors thanks for the revised manuscript. It is improved much more now.

However, checking the Introduction and specifically the references you added, they appear to be old (for example ref 4, 5, 10, 11) they are from 1968-2016. Try to add more recent references such as: https://doi.org/10.3390/electronics11060941 ,  https://doi.org/10.3390/electronics12010201, https://doi.org/10.3390/sym14020410 ,  https://doi.org/10.3390/sym15020471 , https://doi.org/10.3390/a16030151 .  

Kindly add these references in the introduction either briefly or with a few sentences reflecting the main idea or the strengths of fuzzy sets/fuzzy logic.

After these amendments, the manuscript will be up-to-date and substantially improved.

Author Response

concern: Dear authors thanks for the revised manuscript. It is improved much more now.

However, checking the Introduction and specifically the references you added, they appear to be old (for example ref 4, 5, 10, 11) they are from 1968-2016. Try to add more recent references such as: https://doi.org/10.3390/electronics11060941 ,  https://doi.org/10.3390/electronics12010201, https://doi.org/10.3390/sym14020410 ,  https://doi.org/10.3390/sym15020471 , https://doi.org/10.3390/a16030151 .  

Kindly add these references in the introduction either briefly or with a few sentences reflecting the main idea or the strengths of fuzzy sets/fuzzy logic.

After these amendments, the manuscript will be up-to-date and substantially improved.

Author action: Thanks for your comment. We have updated the references and added the ideas of new references, and marked red in Introduction and References.

Reviewer 2 Report

The paper is well-revised, and I am recommending it for publication.

Author Response

Thanks for your consideration.